# From One to Zero: Causal Zero-Shot Neural Architecture Search by Intrinsic One-Shot Interventional Information

## Abstract

"Zero-shot" neural architecture search (ZNAS) is key to achieving real-time neural architecture search. ZNAS comes from "one-shot" neural architecture search but searches in a weight-agnostic supernet and consequently largely reduce the search cost. However, the weight parameters are agnostic in the zero-shot NAS and none of the previous methods try to explain it. We question whether there exists a way to unify the one-shot and zero-shot experiences for interpreting the agnostic weight messages. To answer this question, we propose a causal definition for "zero-shot NAS" and facilitate it with interventional data from "one-shot" knowledge. The experiments on the standard NAS-bench-201 and CIFAR-10 benchmarks demonstrate a breakthrough of search cost which requires merely **8 GPU seconds on CIFAR-10** while maintaining competitive precision.

## 1 Introduction

Neural architecture search has been an interesting topic in the AutoML community [27]. Traditional methods search by training the distinct neural architecture iteratively [31] whose training cost is huge. One-shot model cleverly use a supernet to merge all the singular neural architectures into one and consequently, the waste of search time is largely saved [16]. Further, the gradient-based one-shot method [12] is proposed which acquires robust results on NASNet [32]. Though the one-shot model largely reduces the search cost, it still suffers from a weight-sharing problem, and especially, gradient-based approaches cause degenerate architectures [29]. The work [25] gives theoretical proof for this and subtly uses a progressive tuning metric to discretize the one-shot supernet iteratively which gets awesome neural architectures. However, it still gets degenerate architectures with different training settings.

The brilliant work [5] from Google Brain gives a hint for searching neural networks without tuning the parameters. "To produce architectures that themselves encode solutions, the importance of weights must be minimized". In this manner, a zero-shot neural architecture search (ZNAS) is born. The work [10] firsts propose the idea of ZNAS to be "it does not optimize network parameters during search ". From a one-shot perspective, the "zero-shot" is given credit by "one-shot" where single neural architectures are supposed to be selected from the weight-agnostic supernet [5]. Considering causal weight messages, the "zero-shot" select neural architecture with the minimum impact of any weight parameter [5]. Thus a causal definition is supposed to be that the weight messages are multi-environmentally distributed. Compared to one-shot NAS, zero-shot NAS gets imperfect weight messages due to random initialization and searching without training [10, 2].

A training-free approach is first proposed by the work [13]. Different from the previous zero-shot model [10], the work [13] samples well-trained architectures and get validation accuracy to train the

statistical proxy before it searches. The work [2] follows the way of the previous work [10] and uses the DARTS search space to conduct zero-shot NAS on CIFAR-10 and ImagNet in a training-free manner. However, the number of samples directly decides the belief of the final precision. The "well-trained" architectures might not be "perfectly-trained" in different training settings.

Zero-shot NAS learns the representation of neural architectures to get the best one. Consistently compared to one-shot NAS methods, zeros-shot NAS methods ignore the weight information. By merely measuring the architectural expressivity, they overlooked the impact of weights as a necessary assessment element. From a one-shot NAS perspective, architectural information can be represented by a list of neuron representations [25]. The message of training weights $\omega$ supports the neuron's representation [15, 12, 25]. Because the structural dependencies of shared (mutual) messages across neurons are all agnostic [5], in the zero-shot neural architecture search, the neuron's representation is harder to interpret due to the random messages. What is worse, the uniterpretability might result in large bias and variances because the imprecise observational data might be misleading. Finally, it will lead the search to get degenerate architectures through the process of accumulating errors.

We first propose to interpret the zero-shot NAS in a causal-representation-learning setting. According to the weight-agnostic setting, we formulate the zero-shot NAS as a novel framework for imperfect-information NAS. The structural information of zero-shot NAS is interpreted by impact with the latent factors. As a consequence, intrinsic high-level interventional data acquired by one-shot NAS is properly adopted to refine the imperfectness. Moreover, we reformulate the causality by game theory and interpret the imperfect-information NAS as imperfect information game $\mathcal{G}$. Extensive experiments on various benchmark datasets including CIFAR-10, NAS-Bench-201, and ImageNet have shown the super search efficiency ($10000\times$ faster than DARTS) of our methods. In this work, our main contributions are as follows:

- We propose that the causal zero-shot NAS is to learn the neuron's representation with latent factors in observationally imperfect messages.
- We theoretically demonstrate the validation information of either a neuron or a neuron ensemble obeys a Gaussian distribution given a Gaussian input.
- The proposed method uses high-level interventional data from one-shot NAS to facilitating zero-shot NAS to solve the imperfectness.
- Our method sets the new state-of-the-art in zero-shot NAS of search cost (8 GPU seconds) while maintaining comparable test accuracies.

## 2 Preliminaries and Related Work

In this section, we talk about the preliminaries and the previous works on one-shot NAS and zero-shot NAS. We talk about the motivation to replace statistical proxy by introducing the basic knowledge on causal interventaional representation learning in causality [20, 1].

### 2.1 One-shot NAS

One-shot NAS methods [12, 16], that unify all the single-path neural architectures into one super-network $\mathcal{S}$ (supernet), select the single-path neural architecture as the best one by training the weights $\omega$ in a weight-sharing manner and maximizing the validation accuracy ($\mathcal{V}$) of architecture $\mathcal{A}$ as follows:

$$Max_{\mathcal{A}}(\mathcal{V}(\mathcal{A}, \bar{\omega})) \quad s.t. \quad \bar{\omega} = \omega + \delta_{\mathcal{A}}\omega_{\mathcal{S}} \tag{1}$$

The iterative updating of $\omega$ and selection of $\mathcal{A}$ makes the one-shot NAS a bi-level optimization problem that is NP-hard. Differentiable one-shot model also relies on the observational data from unitedly trained validation accuracies of differentiable subnets [12]. Wang et al. [25] propose a selection-based approach to modify the output of differentiable one-shot NAS [12] to discretize a single-path neural architecture that consists of operations (neurons) with strength. As a consequence, the perturbation-based inductive bias is demonstrated to be helpful to solve the degeneration.

### 2.2 Statistical proxies in zero-shot NAS

We compare the various training-free and zero-shot NAS methods according to the usage of statistical representation. Some training-free approaches use the statistic of validation accuracy to predict the

final architecture. NASWOT [13] samples a number ($N$) of well-trained neural architectures from the NAS-Bench-201 dataset to learn the kernel. However, to get these representations, the training costs tremendously. The zero-shot methods directly use zero-cost statistical proxies to represent the expressivity without weights and validation accuracy. Zen-NAS [10] uses a Gaussian complexity to measure the network expressivity and evolve the architectures to maximize the expressivity. Other training-free approaches such as TE-NAS [2] and NASI [22] imitate the train of NAS by neural tangent kernel (NTK) which largely reduces the waste of train cost. TE-NAS [2] propose to maximize the number of linear region of activation patterns [14]. On the opposite, NASI [22] subtly optimize the trace of NTK by sampling.

Here raise the question that to what extent the validation accuracy outperforms the statistical proxy. Vice versa, we question if the statistical proxy is in substitute of the validation accuracy. Compared to the proxy-based methods with approximations, the validation-based method is more reproducible. The validation accuracy is an intrinsic robust and upper-bounded proxy to measure the neural architectures. Besides, previous arts of one-shot manner usually use the validation accuracy to be the objective to maximize. Despite these benefits, the zero-shot representation is imperfect due to the weight-agnostic messages.

## 2.3 Causal representation learning

The study [20] demonstrates that causality is a "subtle concept" which can not be fully described by Boolean or Probabilistic. It is more about reasoning. Reichenbach demonstrates a common cause principle to explain the causality by dependencies among variables [18]. Causal representation learning mainly deals with learning causally for representations. By observational data, we can hardly learn the real circumstances (environments), especially in complex scenes and high-dimensional data scenarios. Causal representation learning seeks to extract high-level information (dependencies) from low-level data. Interventions have taken a prominent role in representation learning literature on causation. The work [1] uses interventional data to facilitate the causal representations to get precise outcomes. Neural architecture search aims at learning the architectural representations automatically. The automatism of the previous arts of neural architecture search might not be causal especially in zero-shot setting.

# 3 Method

## 3.1 Imperfect information

Neural architecture search is a task aiming at interpreting the mechanism of architectural knowledge of neural networks given methods of evaluations. Activation patterns, statistical proxies, and naive validation accuracy are adopted to evaluate the score of a neural network. However, we can hardly understand any neural network and even hardly explain the weight distribution of any neural network without assumptions. Observational data are always imperfect due to the infinite environments (search spaces/training schemes/hardware/etc.) of all possible networks with finite observations and limited tools. Architecture information is not stand-alone.

In one-shot NAS, demonstrated in Equation 4, given a neural network, we first train the weights $\omega$ and the $\omega$ combined with architecture $\mathcal{A}$ can give a validation accuracy $\mathcal{V}$. After $\mathcal{V}$ is given, we then update the $\omega$ to get $\bar{\omega}$ and a novel architecture $\mathcal{A}$ until the validation accuracy $\mathcal{V}$ is maximum. In the train, the architecture of a neural network is the key factor that impacts the other two factors $\omega$ and validation accuracy $\mathcal{V}$. The search is actually a reverse way of train to the aspect of the intrinsic dependency of accuracy $\mathcal{V}$ on the weight $\omega$ and architecture $\mathcal{A}$. However, we have overlooked a lot of factors like data distributions, batch sizes, rates of weight decay, and so on and on which we can not optimize as "one shot". If the observational data alone can not interpret the phenomenon, it is a must to model the latent factors $\mathcal{Z}$ that cause this uninterpretability. Figure 1 illustrates the dependencies of architecture $\mathcal{A}$, validation accuracy $\mathcal{V}$, and weights $\omega$. The dashed line reveals that $\mathcal{Z}$ changes the dependencies of selected neurons (or searched architectures) on observational data of $\omega$ and $\mathcal{V}$ [23], which indeed implies strong causality [20]. In logical condition, the structural relationship between $\mathcal{V}$ and $\omega$ can be almost broken[1].

---

[1]See demonstration in Section 3.3, results in Section 4.

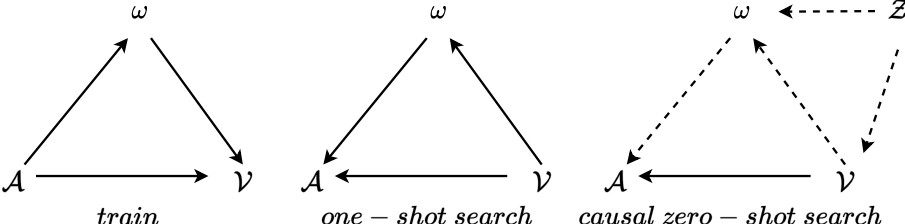

Figure 1: Illustrations of the dependencies of architecture $\mathcal{A}$, validation accuracy $\mathcal{V}$, and weights $\omega$ with latent factor $\mathcal{Z}$ on the train (left), one-shot neural architecture search (middle), and causal zero-shot neural architecture search (right).

We assume the validation accuracy $\mathcal{V}$ of a set of neural architectures $\{\mathcal{A}\}$ obeys a Gaussian distribution.

$$\mathcal{V} \sim \mathcal{N}(\mu, \sigma^2) \tag{2}$$

Due to the random weight information, artificial neural networks (ANN) themselves have architectural information to deliver the neural networks' expressivity with large variances [5]. It is demonstrated that the weight-agnostic neural network still preserves the 92% accuracy-level information for digit classification by the work [5]. However, the weights are agnostic and consequently the validation accuracies are imperfect. We assume the true validation accuracy is the difference of the observational $\mathcal{V}^{obser}$ and latent impact of factor $\mathcal{Z}$ demonstrated in Equation 3.

$$\mathcal{V} \sim \mathcal{N}(\mu_{obser} - \mu_{\mathcal{Z}}, \sigma^2_{obser} - \sigma^2_{\mathcal{Z}}) \tag{3}$$

## 3.2 Problem formulation

In Zen-NAS, the adoption of statistical proxy on the feature map is impressive while it is constrained to structural dependencies [10]. We question to what extent, when we search a neural network, the statistical proxies can be replaced with the more robust functions such as validation accuracy causally [20]. In some one-shot [16, 12] and training-free methods [13], the evaluation metrics are usually the validation accuracy of the associated neural architectures.

Inspired by the previous work [25], we evaluate each neuron to select respectively in substitute. Intuitively, we measure the importance of each neuron by a simple validation accuracy of a singular associate neuron while resting other neurons on the same edge. DARTS+PT [25] the perturbation-based approach mutes the irrelevant neurons to conduct an inference while saving the other paralleled edges. For each paralleled edge (layer) $\mathcal{E}$ that contains $M$ neurons $\mathcal{N}$s, we mute the other neurons while only saving the $i^{th}$ neuron $\mathcal{N}_{(i)}$. The $k^{th}$ paralleled edge $\mathcal{E}_i^{(k)}$ consequently only contains one neuron (operation): $\mathcal{E}_i^{(k)} = \{0 \times \mathcal{N}_{(1)}, 0 \times \mathcal{N}_{(2)}, \ldots, \mathcal{N}_{(i)}, \ldots, 0 \times \mathcal{N}_{(M)}\}$. When saving the other paralleled edges $\{\mathcal{E}_{(j)}\}_{j \neq k}$, $\mathcal{N}_{(i)}$ denotes any single sub-architecture (a neuron) in the supernet $\mathcal{S}$ with tuned weights $\omega_{\mathcal{S}}$ of the supernet. Formally, the one-shot neuron selection for $k_{th}$ paralleled edge is defined as:

$$\mathcal{N}^* = argmax(\mathscr{F}(\{\mathcal{V}(\mathcal{N}_{(i)}, \omega_{\mathcal{S}})\})) \quad \forall \mathcal{N}_{(i)} \in \mathcal{E}^{(k)} \tag{4}$$

where validation accuracy $\mathcal{V}$ is measured by an intrinsic inductive bias function $\mathscr{F}$ such as a reinforcement learning policy $\pi$ [31, 32]. $\mathcal{V}(\mathcal{N}_{(i)}) = \mathcal{V}(\{\mathcal{E}^{(1)}, \mathcal{E}^{(2)}, \ldots, \mathcal{E}_i^{(k)}, \ldots, \mathcal{E}^{(N)}\})$ in practise.

In zero-shot NAS, the weight information is agnostic, which is impacted by a latent factor $\mathcal{Z}$ as shown in Figure 1. [4]. The latent variable obeys a distribution $\mathcal{P}$ in dimension $\Lambda$:

$$\mathcal{Z} \sim \mathcal{P}^{\Lambda} \tag{5}$$

When we sample larger enough numbers of impacts, the sample of factor $\mathcal{Z}$ obeys a Gaussian distribution by the central limit theorem (CLT). The causal zero-shot neural architecture search (Causal-Znas) that searches in imperfect messages is defined as:

$$\mathcal{N}^* = argmax(\mathscr{F}(\{\mathcal{V}(\mathcal{N}_{(i)}, \omega)\}|\mathcal{Z})) \quad \forall \mathcal{N}_{(i)} \in \mathcal{E}^{(k)} \tag{6}$$

for $i = 1, 2, \ldots, M$. In this Equation 6, $Z$ means the latent information to impact agnostic-weights (such as a random initialization [5, 10]) and consequently validation accuracies $\mathcal{V}$. Therefore, we get a causal information set of singular neuron representation $\{\mathcal{V}(\mathcal{N}_{(i)})|\mathcal{Z}\}$ for $i = 1, 2, \ldots, M$. For each paralleled edge (layer) $\mathscr{E}$ that contains $M$ neurons $\mathcal{N}$s: $\mathscr{E} = \{\mathcal{N}_{(1)}, \mathcal{N}_{(2)}, \ldots, \mathcal{N}_{(M)}\}$. We calculate the information of singular neuron $\mathcal{N}_i$ on edge $\mathscr{E}^{(j)}$ by freezing the other layers (ensembles/edges) $\{\mathscr{E}^{(k)}\}_{k \neq j}$ so that the causal information is only impacted by the current neurons due to the same condition (in the same paralleled edge). Then the causal information set of a paralleled edge $\mathscr{E}$ is as:

$$\{\mathcal{V}(\mathscr{E})|\mathcal{Z}\} = \{\mathcal{N}_{(1)}(\mathcal{X}|\mathcal{Z}), \mathcal{N}_{(2)}(\mathcal{X}|\mathcal{Z}), \ldots, \mathcal{N}_{(M)}(\mathcal{X}|\mathcal{Z})\} \tag{7}$$

In a Causal-Znas, a prediction function $\mathscr{F}$ is able to measure the selected architectures from the un-trained supernet. To avoid the improper introduction of inductive biases, we use an identity function to measure the importance of neurons.

### 3.3 Gaussian intervention

Most existing NAS approaches use observational data and make assumptions on the architectural dependencies to achieve provable representation identification. However, in our causal zero-shot neural architecture search, there is a wealth of interventional data available. To perfect the observational validation accuracies $\mathcal{V}^{obser}$ in $\mathcal{D}$, we sample $\mathcal{V}^{ven}$ from an interventional distribution $\mathcal{D}(\mathcal{Z})$ to be in substitute for the ones derived by the observation $\mathcal{V}^{obser}$. Formally, we have: $\mathcal{V}^{ven} \sim \mathcal{D}(\mathcal{Z})$. Though pure architectural information is imperfectly obseved, we can use an interventional function $\mathcal{I}$ (**do intervn** [1]) to replenish data from a one-shot perspective:

$$\mathcal{V} = \mathcal{I}_p^{\mathcal{D}(\mathcal{Z})}\mathcal{V}^{ven} \bigcup \mathcal{I}_{1-p}^{\mathcal{D}}\mathcal{V}^{obser} \tag{8}$$

Ming et al. [10] assume the inputs obey Gaussian distribution and get comparable results with one-shot methods [12, 16]. What we use as the input for each neuron is a Gaussian image which also obeys the assumption of Gaussian inputs of Zen-NAS [10].

**Lemma 1.** *Given a Gaussian input $\mathcal{X} \sim \mathcal{N}(\mu, \sigma^2)$, the output of a neuron $\mathcal{N}$ in the first layer is Gaussian.*

*Proof.* Assuming each neuron is a distinct convolution denoted as $Conv_i$ for $i = 1, 2, \ldots, M$, then the output of this edge is:

$$\mathcal{O} = \sum_{i=1}^{M}(\{Conv_{(1)}(\mathcal{X}, \mathcal{W}_{(1)}), Conv_{(2)}(\mathcal{X}, \mathcal{W}_{(2)}), \ldots, Conv_{(M)}(\mathcal{X}, \mathcal{W}_{(M)})\}) \tag{9}$$

where $\mathcal{X} \sim \mathcal{N}(\mu, \sigma^2)$ and $\mathcal{W}_{(i)} \sim \mathcal{N}(\mu_w, \sigma_w^2)$ for $i = 1, 2, \ldots, M$. Given the i.i.d. inputs and weights, the output score (validation accuracy) of the neural network layer is Gaussian since the Convolution of a Gaussian (random variable) is still a Gaussian (random variable). We have Gaussian weights $\mathcal{W}_{(i)}$ and $Conv_{(i)}(\mathcal{X}, \mathcal{W}_{(i)}) \sim \mathcal{N}(\mu_{(i)}, \sigma_{(i)}^2)$. Then $\sum_i Conv_{(i)}(\mathcal{X}, \mathcal{W}_{(i)}) \sim \mathcal{N}(\sum \mu_{(i)}, \sum \sigma_{(i)}^2)$. $\square$

**Lemma 2.** *Given a Gaussian input $\mathcal{X} \sim \mathcal{N}(\mu, \sigma^2)$, the output of a neuron $\mathcal{N}$ in any layer is Gaussian.*

*Proof.* Apparently, any weighted summation of random variables that obey two distinct Gaussian is still a Gaussian. In neural networks, the layers are stacked. Based on Lemma 1, in the latter layer, the outputs also obey the Gaussian, whose inputs are the former layer's outputs. The convolution (neuron) $Conv'_{(i)}$ of the next layer with output of latter layer $\mathcal{O}$ (in Equation 9) has $Conv'_{(i)}(\mathcal{O}) \sim \mathcal{N}(\mu'_{(i)}, \sigma'_{(i)}{}^2)$. $\square$

**Corollary 2.1.** *Given a Gaussian input $\mathcal{X} \sim \mathcal{N}(\mu, \sigma^2)$, the output of any neuron ensemble $\{\mathcal{N}_{(i)}\}_{i \in \mathcal{M}}$ is Gaussian.*

Formally, we have $\mathcal{O}^{(i)} \sim \mathcal{N}^{(i)}(\mu', \sigma'^2)$. $\widetilde{\mathcal{O}} = \{\mathcal{O}^{(1)}, \mathcal{O}^{(2)}, \ldots, \mathcal{O}^{(K)}\}$ where $\widetilde{\mathcal{O}}$ denotes all the outputs across edges $\overbrace{\mathscr{E}_{(1)}, \mathscr{E}_{(2)}, \ldots, \mathscr{E}_{(K)}}$. Based on Lemma 1 and Lemma 2, we get the Corollary 2.1 to select edges (topology preferences).

207 *Proof.* By Lemma 1, we have any neuron $\mathcal{N}_{(i)}$ has a Gaussian output $\mathcal{O}^{(i)} \sim \mathcal{N}(\mu_{(i)}, \sigma_{(i)}^2)$. Any

208 ensemble of neurons has an output $\sum_i \mathcal{O}^{(i)}$. Then we have $\sum_i \mathcal{O}^{(i)} \sim \mathcal{N}(\sum \mu_{(i)}, \sum \sigma_{(i)}^2)$. $\qquad \square$

209 As demonstrated in Equation 8, we propose an intervention function $\mathcal{I}^{\mathcal{D}}$ to facilitate the imper-
210 fect causal representation of the validation information. We propose that the ideal information is
211 distributed in the information set by a probability $p$. The distribution $\mathcal{D}$ is $\mathcal{N}(\mu, \sigma)$ in the context.

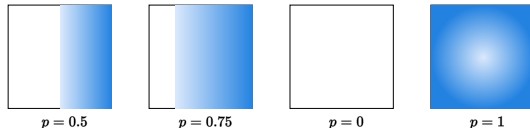

Figure 2: Illustration of intervention of observational data. The blue denotes interventional data while the white denotes observational data.

212 Herein, we question to what extent, the imperfectness can be interventionally refined [1]. We use
213 the parameter $p$ to asymmetrically flipping the random Gaussian $\mathcal{I}_p^{\mathcal{N}(\mu, \sigma^2)}$ [15] to understand the
214 imperfect information in dimension $\Lambda$ which is mapped to a vanilla Gaussian (in Equation 5). As
215 shown in Figure 2, it compares the information difference between the observational information set
216 and interventional information set impacted by the parameter $p$. In different environments, the data
217 of interventional data combined with observation obeys a distinct Gaussian, which implies strong
218 coherence and robustness. When $p = 1$, the causality is perfectly achieved due to breaking the
219 dependency of validation accuracy $\mathcal{V}$ on weights $\omega$; otherwise, it is imperfect. The mean and variance
220 coefficients of the additional notion of intervention are derived by sampling validation accuracy of
221 one-shot prior. We propose that setting of $p$ is conditional on the fraction of the mean of latent factor
222 to the difference of the mean of observational data and the mean of interventional data.

223 **Proposition 1.** *When* $p \longrightarrow \frac{\mu_{\mathcal{Z}}}{\mu_{obser} - \mu_{ven}}$*, the mean of the intervened data* $\widetilde{\mu} \longrightarrow \mu_{true}$*.*

224 As demonstrated in Proposition 1, a sufficient condition of the mean of intervened data is getting
225 closer to the true mean of the validation accuracy is that the $p$ is closer to 1 and interventional data is
226 closer to the true data.

227 ### 3.4 Causal zero-shot neural architecture search

228 We formulate the zero-shot NAS into ensemble selection and neuron selection. There are $K$ neuron

229 ensembles $\overbrace{\{\mathcal{N}_{(i)}\}_{i \in \mathcal{M}}^{(1)}, \{\mathcal{N}_{(i)}\}_{i \in \mathcal{M}}^{(2)}, \dots, \{\mathcal{N}_{(i)}\}_{i \in \mathcal{M}}^{(K)}}$. For each ensemble, there are $M$ neurons

230 (operations). The ensemble selection is the selection of an ensemble $\{\mathcal{N}_{(i)}\}_{i \in \mathcal{M}}^{(j)}$ of neurons among

231 the $K$ ensembles ($j \in \mathcal{K}$), while neuron selection follows the same formula and selects a neuron $\mathcal{N}_{(i)}$

232 from a neuron ensemble $\{\mathcal{N}_{(i)}\}_{i \in \mathcal{M}}^{(j)}$.

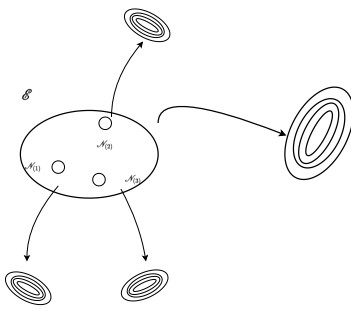

Figure 3: The distribution plate of three neurons and a big distribution plate of ensemble of them.

**Algorithm 1** Causal zero-shot neuron selection.

---

Initialize supernet weights $\omega$;
For $i = 1, 2, \ldots, M$:
    Calculate validate accuracy $\mathcal{V}^{obser}(\mathscr{N}_{(i)}(\omega))\}$;
    **do intervn** by $p$;
Maximize the $\mathcal{V}$ and select the $\mathscr{N}^*$.

---

As is shown in Figure 3, the validation accuracy of both a neuron and a neuron ensemble obey Gaussian distributions respectively. From a macro perspective it is an ensemble selection while from a minor perspective, it is a neuron selection. Thus we talk about both types in the same formula.

As demonstrated in Equation 6, the final outcome neurons are derived by maximizing their validation accuracies according to the latent factor. Given the Gaussian intervention in Equation 8, we further modify the formula of the causal neuron selection by doing intervention (without the additional inductive bias [20]):

$$\widetilde{\mathscr{N}^*} = argmax(\{\widetilde{\mathcal{V}}(\mathscr{N}_{(i)})\}_{i \in \mathcal{M}}) \tag{10}$$

, where $\widetilde{\mathcal{V}}$ is the validation accuracy with intervention.

The methodology of neuron selection is given in Algorithm 1. The search process of neuron ensemble follows the same formulation as mentioned in this Section. **do intervn** represents to do intervention. At first, the weight $\omega$ of the supernet is randomly initialized [10]. Second, validation scores $\mathcal{V}$ on the validation set are prepared for the calculation of the neurons $\mathscr{N}$ which adopts probability $p$ to do the intervention. At last, the maximum of values is compared to select the best neuron (operation). In practice, when the probability $p$ is close to 1, the validation accuracy of observation has less need to compute.

Equation 6 reveals a universal formula for causal neural architecture search in the zero-shot settings. The measure function $\mathscr{F}$ measures the importance [25] ("responsibility") of a neuron and Shapley value is proposed to be ideal for the selection of a neuron [7] or ensemble [19].

$$\mathscr{N}^* = argmax(\{\mathcal{G}_{(i)}(\{\widetilde{\mathcal{V}}\})\}_{i \in \mathcal{M}}) \tag{11}$$

We use the game-theoretic inductive bias to extract the valuable information [20, 7]. $\mathcal{G}$ represent the Shapely value [21]. Given Corollary 2.1, we know that any the neuron ensemble obeys a Gaussian distribution. The information set of Shapley value is thus build on top of an ensemble of Gaussian variables. However, we could not guarantee a Gaussian distribution of the Shapley value [24]. As a consequence, we use a Gaussian distribution to do intervention on validation accuracy and then calculate the Shapely value of the intervened validation accuracy. At last, the Shapley value is maximized whose associated neuron is supposed to be more expressive [7].

## 3.5 Weight-agnostic weights

In the assumptions of various methods, weights are initialized as Gaussian. However, in our framework, we demonstrate that this strong assumption is not a must. Supernet can be initialized in different ways: i) with Gaussian [10], ii) Uniform [5], and iii) Constant number [5].

**Corollary 2.2.** *Given a Gaussian input $\mathcal{X} \sim \mathcal{N}(\mu, \sigma^2)$, if the initial weights are Uniform or Constant number $C$, the output of any neuron ensemble $\{\mathscr{N}_{(i)}\}_{i \in \mathcal{M}}$ is not Gaussian.*

*Proof.* Apparently, the convolution of a Gaussian input with constant or uniform weights obeys a difference of CDF $\Phi$ of the Gaussian in the range of constant or uniform. □

In the previous work [5], it is proposed that weights are supposed to be initialized by a distribution but not a constant ($C$). To be more precise, we propose that the constant value could not represent the agnostic weights and thus could not reflect the latent information while a uniform distribution can guarantee the randomness. By training on a "wide range" of uniform weight samples, Gaier et al. propose that "the best performing values were outside of this training set" [5]. We propose that this phenomenon is essentially resulted from a distribution shift of the Gaussian validation accuracy which causes the change of search procedure. To solve the distribution shift, we could use the difference of CDF of Gaussian ($\Phi$) to conduct intervention. Even in a broader view, if the weights distributions are totally unknown, we can use Bayesian method to approximate a distribution $\mathcal{D}(\mathcal{Z})$ in Equation 8.

# 4 Experiments

We present the results and all experiment details of our method in this section. A robustness analysis is included to examine the stability of our method, which also explains the time efficiency. Results are given on the benchmark datasets, NAS-Bench-201 and CIFAR-10.

## 4.1 Experimental details

We use the search space of DARTS [12] for fair comparisons with the state-of-the-art NAS approaches. During the searching process, we follow adopting the **same** and hyper-parameters as DARTS [12] to initialize the supernet on the CIFAR-10 and NAS-Bench-201 datasets for a fair comparison with DARTS-variants (one-shot methods). All the training is conducted on a single 2080Ti GPU.

## 4.2 Results on CIFAR-10

Table 1: Comparison with state-of-the-art NAS methods on CIFAR-10.

| Algorithm | Test Error (%) | Params (M) | Search Cost (GPU seconds) | Search Strategy |
|---|---|---|---|---|
| DenseNet-BC [6] | 3.46 | 25.6 | - | manual |
| NASNet-A + cutout [32] | 2.65 | 3.3 | $1.73 \times 10^8$ | RL |
| AmoebaNet-A [17] | $3.34 \pm 0.06$ | 3.2 | $2.72 \times 10^8$ | GA |
| AmoebaNet-B [17] | $2.55 \pm 0.05$ | 2.8 | $2.72 \times 10^8$ | GA |
| PNAS [11] | $3.41 \pm 0.09$ | 3.2 | $1.94 \times 10^7$ | SMBO |
| ENAS [16] | 2.89 | 4.6 | 43200 | RL |
| DARTS(1st) [12] | $3.00 \pm 0.14$ | 3.3 | 34560 | gradient |
| DARTS(2nd) [12] | $2.76 \pm 0.09$ | 3.3 | 86400 | gradient |
| BayesNAS [30] | $2.81 \pm 0.04$ | 3.4 | 17280 | gradient |
| DrNAS [3] | $\mathbf{2.54 \pm 0.03}$ | 4.0 | 34560 | gradient |
| ISTA-NAS [26] | $2.54 \pm 0.05$ | 3.3 | 4320 | gradient |
| DARTS+PT [25] | $2.61 \pm 0.10$ | 3.0 | 69120 | gradient |
| TE-NAS [2] | $2.63 \pm 0.06$ | 3.8 | 4320 | NTK |
| NASI-FIX [22] | $2.79 \pm 0.01$ | 3.9 | 864 | NTK |
| NASI-ADA [22] | $2.90 \pm 0.01$ | 3.7 | 864 | NTK |
| Causal-Znas($p = 0.5$) | $2.89 \pm 0.08$ | **2.6** | 142 | causal |
| Causal-Znas($p = 1$) | $2.75 \pm 0.10$ | 3.2 | **8** | causal |
| Causal-Znas-G($p = 1$) | $2.61 \pm 0.04$ | 3.1 | 30 | causal |

As shown in Table 1, we compare the proposed Causal-Znas and game-version Causal-Znas-G with the state-of-the-art methods. The comparisons are made with respect to the informatics of the model, including test accuracy on the test set (Test Error), the number of parameters (Params), the search costs, and the search strategies. As shown, our results set the new state-of-the-art search speed with a competitive test error rate. Compared to DARTS [12], our method is 10000× faster with comparable accuracy (2.75% v.s. 2.76%). Compared to DARTS+PT [25], our model is much simpler without introducing the perturbation-based inductive bias [20] and achieves a similar test error rate (2.61% v.s. 2.61%). DrNAS [3] and ISTA-NAS [26] are not only precise (2.54%) but also theoretically sound approaches. ISTA-NAS [26] is extremely fast in one-shot NAS while ours are more competitive (500× faster) in search efficiency.

We compare our method with other zero-shot NAS approaches in Table 1. It demonstrates that the TE-NAS [2] which is the first algorithm that reaches 4 GPU hours search cost is experimentally awesome. TE-NAS uses the neural tangent kernel to approximate the train so it largely reduces the cost of training the neural networks. Compared to TE-NAS, our proposed approach is 500× faster and our game-based result (-G) gets a comparable test error rate (2.61% v.s. 2.63 %) with a smaller number of parameters (3.1M v.s. 3.8M). We also surpass the current state-of-the-art zero-shot (training-free) method (NASI) [22] by more than 100× in search efficiency and get fewer errors in both settings (2.75% v.s.2.79%; 2.89% v.s. 2.90%).

## 4.3 Results on NAS-Bench-201

NAS-Bench-201 is a pure-architecture-aware dataset where the neural architectures are trained in the same settings, and the info such as performance, parameters, architecture topologies, and operations

are available. Compared to NAS-Bench-101 [28], NAS-Bench-201 adopts a different search space and gets results on various datasets such as CIFAR-10, CIFAR-100, and ImageNet16-120.

As shown in Table 2, it compares our proposed method with the state-of-the-art methods on NAS-Bench-201. Compared to NASWOT(N=10) [13], NASWOT(N=100) and NASWOT(N=1000) are much more accurate due to enlarged sample amounts. However, it also cause $10\times$ and $100\times$ waste of search costs. NASI [22] also enlarges its search cost to get much more precise results with extension of 90s. Our approach gets the same search cost with NASWOT (3s) while being much more precise on CIFAR-10 (90.03% v.s. 89.14%, 93.49% v.s. 92.44), CIFAR-100 (70.18% v.s. 68.50%, 71.18% v.s. 68.62%) and ImageNet 16-120 (43.83% v.s. 41.09%, 44.43% v.s. 41.31). A 9s extension of search cost (**Ours-G**) by neuron games gets even better results than NASWOT and NASI for their extreme results.

Table 2: Comparison with the state-of-the-art methods on NAS-Bench-201.

| Algorithm | Search Cost | CIFAR-10 | | CIFAR-100 | | ImageNet 16-120 | |
|---|---|---|---|---|---|---|---|
| | GPU seconds | Val (%) | Test (%) | Val (%) | Test (%) | Val (%) | Test (%) |
| ResNet [8] | - | 90.83 | 93.97 | 70.42 | 70.86 | 44.53 | 43.63 |
| **Optimal** | - | **91.61** | **94.37** | **73.49** | **73.51** | **46.77** | **47.31** |
| RSPS [9] | 7587 | $84.16 \pm 1.69$ | $87.66 \pm 1.69$ | $45.78 \pm 6.33$ | $46.60 \pm 6.57$ | $31.09 \pm 5.65$ | $30.78 \pm 6.12$ |
| DARTS(1st) [12] | 10890 | $39.77 \pm 0.00$ | $54.30 \pm 0.00$ | $15.03 \pm 0.00$ | $15.61 \pm 0.00$ | $16.43 \pm 0.00$ | $16.32 \pm 0.00$ |
| DARTS(2nd) [12] | 29902 | $39.77 \pm 0.00$ | $54.30 \pm 0.00$ | $15.03 \pm 0.00$ | $15.61 \pm 0.00$ | $16.43 \pm 0.00$ | $16.32 \pm 0.00$ |
| NASWOT(N=10) [13] | **3** | $89.14 \pm 1.14$ | $92.44 \pm 1.13$ | $68.50 \pm 2.03$ | $68.62 \pm 2.04$ | $41.09 \pm 3.97$ | $41.31 \pm 4.11$ |
| NASWOT(N=100) [13] | 30 | $89.55 \pm 0.89$ | $92.81 \pm 0.99$ | $69.35 \pm 1.70$ | $69.48 \pm 1.70$ | $42.81 \pm 3.05$ | $43.10 \pm 3.16$ |
| NASWOT(N=1000) [13] | 300 | $89.69 \pm 0.73$ | $92.96 \pm 0.81$ | $69.86 \pm 1.21$ | $69.98 \pm 1.22$ | $43.95 \pm 2.05$ | $44.44 \pm 2.10$ |
| NASI(T) [22] | 30 | - | $93.08 \pm 0.24$ | - | $69.51 \pm 0.59$ | - | $40.87 \pm 0.85$ |
| NASI(4T) [22] | 120 | - | $93.55 \pm 0.10$ | - | $71.20 \pm 0.14$ | - | $44.84 \pm 1.41$ |
| **Ours** | **3** | $90.03 \pm 0.61$ | $93.49 \pm 0.71$ | $70.18 \pm 1.38$ | $71.18 \pm 1.41$ | $43.83 \pm 2.10$ | $44.43 \pm 2.11$ |
| **Ours-G** | 12 | $90.12 \pm 0.52$ | $93.59 \pm 0.67$ | $70.54 \pm 1.29$ | $71.50 \pm 1.31$ | $45.77 \pm 1.20$ | $45.73 \pm 1.21$ |

## 4.4 Results on ImageNet with the DARTS search space

As shown in Table 3, we report the searched results on ImageNet. The validation size of the observation data batch is 1024. On ImageNet, the number of classes is 1000 so a large data batch is necessary. Compared to NASI [22], and TE-NAS [2], our search costs are faster when $p = 1$. The larger batches for evaluation enlarge the search cost for observational data resulting in a slightly larger search cost when $p = 0.5$. **Ours(p=1)** gets a competitive test error rate (25.0%) in the table and NASI-ADA [22] gets similar result (24.8%) but NASI-ADA has a larger search cost (864s v.s. 8s).

Table 3: Comparisons with the state-of-the-art on ImageNet.

| Algorithm | Search Cost (GPU seconds) | Test Error (%) | Params (M) |
|---|---|---|---|
| DARTS [12] | $8.64 \times 10^5$ | 26.7 | 4.7 |
| DARTS+PT [25] | $2.94 \times 10^5$ | 25.5 | **4.6** |
| DrNAS [3] | $3.37 \times 10^5$ | **24.2** | 5.2 |
| TE-NAS [2] | 4320 | 26.2 | 5.0 |
| TE-NAS [2] | 14688 | 24.5 | 5.4 |
| NASI-ADA [22] | 864 | 24.8 | 5.2 |
| NASI-FIX [22] | 864 | 24.3 | 5.5 |
| **Ours(p=0.5)** | 1020 | 25.5 | 4.9 |
| **Ours(p=1)** | **8** | 25.0 | 5.2 |
| **Ours-G** | 31 | 24.8 | 5.4 |

## 5 Conclusion

In this work, we interpret the zero-shot NAS as a causal representation learning and solve it by interventional data from one-shot NAS. Besides, our work is dedicated to displaying the inheriting relationship among the latent variables. We demonstrate that the neural architectures can be evaluated and selected by a Gaussian distribution given Gaussian inputs. Experiments on benchmark datasets reveal awesome efficiency and competitive accuracy.

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
