# A    The detailed description of Proposition 1

In the Proposition 1 in the main body, we talk about the condition that the intervention gets the perfect mean compared to the real mean. Given Equation 3 in the main body, we have $\mu_{true} = \mu_{obser} - \mu_{\mathcal{Z}}$. Assuming $p^*$ is the optimal probability for flipping and interventional data and observational data, then we have:

$$p^* \mu_{ven} + (1 - p^*)\mu_{obser} = \mu_{true} \tag{1}$$

$$p^* \mu_{ven} + (1 - p^*)\mu_{obser} = \mu_{obser} - \mu_{\mathcal{Z}} \tag{2}$$

$$p^*(\mu_{ven} - \mu_{obser} = -\mu_{\mathcal{Z}} \tag{3}$$

$$p^* = -\frac{\mu_{\mathcal{Z}}}{\mu_{ven} - \mu_{obser}} \tag{4}$$

$$p^* = \frac{\mu_{\mathcal{Z}}}{\mu_{obser} - \mu_{ven}} \tag{5}$$

The obvious condition that satisfies this Equation 5 is that $\mu_{ven}$ is getting closer and closer to the real mean $\mu_{true}$ as well as $p$ is closer to 1. The reason is obvious. Assuming the interventional data is perfect: $\mu_{ven} = \mu_{obser} - \mu_{\mathcal{Z}}$. Then we have $p^* = 1$ according to Equation 5.