# OpenReview forum: "From One to Zero: Causal Zero-Shot Neural Architecture Search by Intrinsic One-Shot Interventional Information"
_NeurIPS.cc/2023/Conference — Submitted to NeurIPS 2023_

### Official Review · Reviewer_CY83 · 2023-07-02

**Soundness:** 2 fair
**Presentation:** 3 good
**Contribution:** 2 fair
**Rating:** 2
**Confidence:** 4

**Summary:**

This paper proposes a 0-shot NAS method. The key point is that there are latent factors that can influence the architecture search procedure, making the validation accuracy of one-shot NAS unreliable. The method adopts Gaussian intervention to the data and evaluates each operation's performance to reduce the bias brought by validation dataset sampling. Some experiment on CIFAR-10, NAS-Bench-201, and ImageNet show that the method can efficiently search architectures.

**Strengths:**

1. The paper is well written and easy to follow.
2. The paper considers the latent factors that may influence the NAS validation, which has been hardly considered in existing works.
3. The proposed method uses causal inference techniques to solve the NAS problem.

**Weaknesses:**

1. Some definitions and connotations are unclear in the paper. See Questions for more details.
2. The proposed algorithm is too straightforward. It seems directly use intervention technique in NAS procedure, but do not consider any characteristics of NAS problems, which makes the contribution of the method limited.
3. In the experiment part, the authors claim the efficiency of the method, but lots of 0-shot NAS methods has a very high efficiency. The authors do not compare with those 0-shot NAS methods in this part.

**Questions:**

1. The paper is inconsistent about the connotation of the latent factors. At Line 127, the paper states that many factors like data distributions, batch sizes, rates of weight decay can affect validation. However, according to Section 3.3, the authors seem to consider only the data distribution as the latent factor. What is the exact connotation of the latent factors considered in your paper?
2. At Line 135, why validation accuracy obeys Gaussian distribution? In my perspective, under some assumptions, the logits can follow a Gaussian distribution. But when it comes to accuracy, it is hard to say it also follows Gaussian Distribution.
3. How to define "true validation accuracy" in Line 140? I tried to understand that the author was trying to say that there is a true distribution of latent factors, and that the results measured in this true distribution are the true accuracy. But how to define the true distribution?

---

> ### Author Rebuttal · Authors · 2023-08-02
>
> Thanks for reviewing this manuscript. Here are some questions:
> 1. What do you mean by 'definitions'? The validation accuracy is the accuracy of the validation set.
> 2. What do you mean by 'straightforward;? As far as I know, straightforwardness is a strength in writing. Besides, our proof of the Gaussian distribution of any zero-shot NAS is significant theoretically.
> 3. We redefine the framework of zero-shot NAS. We have no need to compare with those less efficient works. Ours is the most efficient of all.
>
> Here are some answers to your questions:
> 1. It is a good question, the other factors can be considered but not a must.

---

> > ### Comment · Reviewer_CY83 · 2023-08-19
> > **Response to Rebuttal**
> >
> > Too many unsolved concerns. Thus, I decrease the rating to strong reject.

---

### Official Review · Reviewer_KBjx · 2023-07-05

**Soundness:** 3 good
**Presentation:** 2 fair
**Contribution:** 2 fair
**Rating:** 3
**Confidence:** 4

**Summary:**

This paper formulates zero-shot NAS as a causal-representation-learning. Further, it uses the high-level interventional data from one-shot NAS to facilitate zero-shot NAS to refine the imperfectness. Extensive experiments achieved comparable performance results on multiple benchmarks.

**Strengths:**

1) This paper proposes to use the high-level interventional data to facilitate zero-shot NAS to address the imperfect-information issue.

2) This paper provides theoretical support for the proposed approach.

**Weaknesses:**

1) The novelty is incremental compared to baseline Zen-NAS approach, and also directly applies the Shapley value which is also leveraged in Shapley-NAS.

2) Although the search cost is low, the achieved performance improvement is not significant.

**Questions:**

1) Please provide more details about the differences between this work and Zen-NAS、Shapley-NAS, as described in Weakness 1.

2) The experiments should be compared with the baseline methods, including Zen-NAS and Shapley-NAS, as well as with the latest SOTA zero-shot methods such as ZiCo.

3) What about the ranking consistency in different NAS benchmark?

4) Minor Errors

    a) Line 40: "zeros-shot" -> "zero-shot"

    b) Line 313: "92.44" -> "92.44%"; Line 314: "41.31" -> "41.31%"

**Limitations:**

The novelty is limited, and the achieved performance improvement is limited.

---

> ### Author Rebuttal · Authors · 2023-08-02
>
> Thanks for reviewing this manuscript. Here are some questions:
> 1. We do not use Zen-NAS as the baseline but build it on our own. Please could you explain why our work is related to Zen-NAS? And why 'the novelty is incremental'?
> 2. The search cost is significant in NAS. Have you ever read the related works?

---

> ### Comment · Reviewer_KBjx · 2023-08-20
> **Response to Rebuttal**
>
> None of the issues were addressed. Therefore, I decrease my score to Reject.

---

### Official Review · Reviewer_7wS7 · 2023-07-07

**Soundness:** 2 fair
**Presentation:** 3 good
**Contribution:** 3 good
**Rating:** 3
**Confidence:** 3

**Summary:**

This paper presents a causal definition of zero-shot NAS and facilitate this with interventional one-shot knowledge data. The paper theoretically demonstrates the validation information of either a neuron or a neuron 60 ensemble obeys a Gaussian distribution given a Gaussian input. It then uses high level interventional data from one-shot NAS to solve the imperfectness of zero-shot NAS. The zero-cost NAS method is studied on DARTS space and NAS Bench 201 with very low search cost while maintaining comparable test accuracies.

**Strengths:**

The paper is well-written and novel. Studying NAS from a causality perspective is original and interesting.

**Weaknesses:**

- The main weakness in my opinion is the weak evaluation of the approach. [NAS-Bench Suite Zero](https://arxiv.org/pdf/2210.03230.pdf) provides easy access and evaluation on about 13 proxies and 28 tasks. Since the performance of a given proxy can vary widely depending upon the search space, task and datasets I am not convinced of the effectiveness of the proxy based on only the results on NATS-Bench and DARTS spaces. Furthermore how would one apply this method to transformer spaces (eg [AutoFormer](https://openaccess.thecvf.com/content/ICCV2021/papers/Chen_AutoFormer_Searching_Transformers_for_Visual_Recognition_ICCV_2021_paper.pdf) , [HAT](https://arxiv.org/abs/2005.14187)) and MobileNets(eg: [OFA](https://arxiv.org/pdf/1908.09791.pdf), which prove queriable validation accuracy based on a surrogate. Can this method be applied in these spaces? Since modern one-shot NAS methods are applied on transformer and mobilenet spaces too, it is important to design a proxy that does indeed generalize well. I encourage the authors to evaluate their method on these search spaces too.
- The results in table 1 and table 2 show a drop in accuracy ( though the num params is lower and search cost is low). Hence I am not very convinced of the effectiveness of the method in finding effective architectures.
- Since reproducibility is quite important in NAS, I think it is very important for the authors to release their code (I couldn't find the code attached).
- Correlation coefficient of the ranking with the true ranking is not studied


**Questions:**

Most questions are covered in the weakness part.
My questions are as summarized below:
1. Could the authors evaluate their method on all the tasks in [NAS-Bench Suite Zero](https://arxiv.org/pdf/2210.03230.pdf) and discuss about the applicability of the method to transformer spaces?
2. Could the authors report the correlation coefficient of the predicted ranking with the true ranking ?
3. Would the authors be releasing the code?
4. Could the authors confirm that the best practices [here](https://www.automl.org/nas_checklist.pdf) are followed?


**Limitations:**

I encourage the authors to discuss the limitations of the proposed method more extensively for eg: assumptions which are architecture type specific? search space generality? Any assumptions which may not hold in practice?

---

> ### Author Rebuttal · Authors · 2023-08-02
>
> Thanks for reviewing this manuscript. We will consider the dataset you mentioned. However, there are too many zero-shot neural architecture search datasets. I don't think it is a must. Anyway, thank you for this suggestion.

---

> > ### Comment · Reviewer_7wS7 · 2023-08-16
> >
> > I have read the response and most of my concerns still remain. I keep my score

---

### Official Review · Reviewer_C1bZ · 2023-07-08

**Soundness:** 2 fair
**Presentation:** 2 fair
**Contribution:** 2 fair
**Rating:** 3
**Confidence:** 4

**Summary:**

This paper proposes a causal zero-shot neural architecture search (NAS).  The NAS problem is decomposed into two components: ensemble selection and neuron selection. By employing the Gaussian intervention to approximate validation accuracy, the authors adapt the perturbation-based approach from DARTS+PT to search for architectures. The performance of their proposed Causal-Znas is evaluated on both NAS-Bench-201 and DARTS search spaces.

**Strengths:**

- It is reasonable to consider the one-shot interventional information for zero-shot NAS.

**Weaknesses:**

- The overall contribution of this research appears to be limited. The combination of zero-shot NAS with DARTS+PT is not considered novel and does not provide significant insights.
- There are some omissions of recent related works in Section 2.2, such as GradSign[1] and ZiCo[2].
- The experimental comparison provided is insufficient. The results of Zen-NAS are missing in the comparison, and it would be beneficial to include a comparison with GradSign and ZiCo. Additionally, recent works often conduct experiments on other tasks such as NLP and ASR, which could provide further insights.

[1] Zhihao Zhang and Zhihao Jia. Gradsign: Model performance inference with theoretical insights. In ICLR, 2022.

[2] Guihong Li, Yuedong Yang, Kartikeya Bhardwaj, and Radu Marculescu. Zico: Zero-shot nas via inverse coefficient of variation on gradients. In ICLR, 2023.

**Questions:**

Please see the Weaknesses.

---

> ### Author Rebuttal · Authors · 2023-08-02
>
> Thanks for reviewing this manuscript. I am very pleased that you mentioned some related works in Zeo-shot NAS. Here are some questions:
> 1. Please could you explain the way our work is related to DARTS+PT? We have not found a clue yet.
> 2. Does ZiCo a common theoretical approach or just an incremental one compared to Zen-NAS?

---

### Decision · Program_Chairs · 2023-09-21

**Decision:**

Reject

**Comment:**

The paper introduces Causal-Znas, a causal zero-shot neural architecture search (NAS) method, but faces criticism for weak evaluation and limited novelty. It is tested on NAS-Bench-201 and DARTS spaces, but not on other relevant architectures like transformers and MobileNets. Results show lower accuracy despite reduced parameters and search costs. The paper is also critiqued for not releasing code for reproducibility and not comparing its efficiency with other zero-shot NAS methods. Overall, the contribution is seen as incremental and lacking in thorough evaluation.

All the reviewers and AC were very disappointed by the **unprofessional and impolite response** from the authors, and the rebuttal indeed failed to provide informative clarifications. AC recommends reject.